# Dietary Chitosan Supplementation Improved Egg Production and Antioxidative Function in Laying Breeders

**DOI:** 10.3390/ani12101225

**Published:** 2022-05-10

**Authors:** Yinhao Li, Qingyue Zhang, Yonghui Feng, Sumei Yan, Binlin Shi, Xiaoyu Guo, Yanli Zhao, Yongmei Guo

**Affiliations:** Inner Mongolia Key Laboratory of Animal Nutrition and Feed Science, College of Animal Science, Inner Mongolia Agricultural University, Huhhot 010018, China; yinhaoli2019@126.com (Y.L.); alicezqy@126.com (Q.Z.); fengyonghui0912@163.com (Y.F.); shibinlin@yeah.net (B.S.); gxy_2594@163.com (X.G.); ylzhao2010@163.com (Y.Z.); ymguo2015@163.com (Y.G.)

**Keywords:** chitosan, laying breeders, production performance, antioxidant functions

## Abstract

**Simple Summary:**

Chitosan is a natural, non-toxic and biodegradable compound, which has antibacterial, antioxidant and anti-tumor properties. Several studies have shown that chitosan also improve the antioxidant capacity of poultry. Recent research showed that chitosan decreased oxidative damage by activating the nuclear factor erythroid-2 related factor 2 pathway, then elevated the meat quality of broilers. Egg breeders are susceptible to oxidative stress during peak egg production, which increase their susceptibility to diseases and lead performance decline. In addition, previous reports on the effect of chitosan on poultry production performance were inconsistent. Based on above reports, this study explored whether chitosan could promote the production performance, and antioxidant defense of laying hens by affecting the nuclear factor erythroid-2 related factor 2 pathway. The results showed that addition chitosan to layer hen diet could increase egg production and feed conversion ratio, and the effect was better at the level of 250~500 mg/kg; as well as, chitosan promoted the antioxidant status in serum, liver and duodenum tissues and the effect was better at the level of 500 mg/kg. Chitosan was likely to increase antioxidant enzyme activities by enhancing the expression of nuclear factor erythroid-2 related factor 2, thereby improving the antioxidant capacity of laying breeders.

**Abstract:**

This study was conducted to explore the dietary effect of chitosan on the production performance, and antioxidative enzyme activities and corresponding gene expression in the liver and duodenum of laying breeders. A total of 450 laying breeders (92.44% ± 0.030% of hen-day egg production) were randomly assigned to five dietary treatments fed 8 weeks: maize-soybean meal as the basal control diet and the basal diet containing 250, 500, 1000 and 2000 mg/kg of chitosan, respectively. Each treatment was randomly divided into 6 equal replicates, with 15 laying breeders in each replicate. The results showed that dietary chitosan could increase hen-day egg production and feed conversion ratio, especially at the level of 250~500 mg/kg; however, chitosan had no prominent effect on feed intake and average egg weight. Dietary chitosan could dose-dependently promote the antioxidant status in serum, liver and duodenum of layer breeders. It has a better promotion effect at the level of 500 mg/kg; however, the effect was weakened at the level of 2000 mg/kg. Chitosan was likely to enhance the gene expression and activities of Nrf2-mediated phase II detoxification enzyme by up-regulating the expression of *Nrf2*, thereby improving the antioxidant capacity of laying breeder hens.

## 1. Introduction

The accumulating oxidative damage in laying hens increase the susceptibility to various diseases, leading to a decline of performance and even resulting in death [1]. Modulating dietary treatments of livestock such as adding antioxidants is one of the effective means to relieve oxidative stress potentials among various methods [2]. Chitosan is a natural source of alkaline polysaccharides, which is a deacetylated form of chitin, mainly found in the exoskeletons of shrimps, crabs and insects [3]. It has advantages of non-toxicity and biocompatible [4], and has functions of anti-oxidation, anti-tumor and immune regulation [5]. It is also renewable and cheap. Thus, chitosan can be effective as a pro-health feed supplement for farm animals, as well as an alternative to feed antibiotics.

It has been proved that chitosan has strong antioxidant capacity in vitro [6]. It is beneficial to antioxidant activity of pancreatic islet cells (NIT-1 cell line) from Mus Musculus (mice) in vitro [7]. As well as, it could reverse the decrease of glutathione peroxidase (GSH-Px) and catalase (CAT) activity, meanwhile, reversed the increase of malondialdehyde (MDA) levels in liver cells from LPS-induced mice [8]. Moreover, chitosan could reverse the decrease of GSH-Px, CAT and total superoxide dismutase (T-SOD) activity, and the increase of MDA level in serum from weaned piglet with intraperitoneal injection of Diquat [9]. Several studies have shown that chitosan can also improve the antioxidant capacity of poultry. Supplementation of chitosan to the diet enhanced the activity of ileal mucosa antioxidant enzyme activity in Gallus gallus (Arbour Acres broilers) [10,11], and improved the fracture strength, bending load and mineralization of femur in ISA Brown laying hens [12].

Nuclear factor erythroid-2 related factor 2 (Nrf2) can regulate the relative expression of antioxidant enzymes, providing defense for the body’s antioxidant function. Under normal conditions, Nrf2 is in a non-free state, avoiding the sensitivity of cells to stimulus. When cells are under oxidative stress, free Nrf2 enters the nucleus and forms a heterodimer with specific proteins, which recognize antioxidant response elements (ARE), thereby regulating the expression of Nrf2-mediated phase II detoxifying enzyme genes, such as *GSH-Px*, *superoxide dismutase* (*SOD*), *CAT* and *heme oxygenase-1* (*HO-1*) [13]. As an effective stimulus of Nrf2, nutritional intervention could repair Nrf2 level, then regulated phase II detoxifying enzyme gene expression, and changed the antioxidant defense capacity [14,15]. Recent research showed that supplementation of chitosan oligosaccharides to diet could decrease ROS production, activating the Nrf2 pathway and Nrf2-mediated *GSH-Px* and *HO-1* gene expression, then elevated the meat quality of broilers exposed to acute heat stress [16]. Cheng et al. (2022) also found that diets supplemented with rare earth chitosan chelate enhanced the activity of antioxidant enzymes in broiler liver by up-regulating the gene expression of the Nrf2 pathway [17]. It is speculated that chitosan may regulate the antioxidant capacity of laying hens through the Nrf2 pathway to adjust the expression of Nrf2-mediated phase II detoxifying enzymes.

In addition, previous reports on the effect of chitosan on poultry production performance are inconsistent. Shi (2005) found that diets containing 0.05–0.1% chitosan achieved better results on both body weight gain (BWG) and feed conversion ratio (FCR) [18]. Khambualai et al. (2009) found that diet containing 0.06% chitosan improved the average feed intake (AFI) and BWG [19]. Diet supplemented with chitosan could increase hen-day egg production (HDEP) and daily egg mass, decrease the content of cholesterol in yolks [20] and increase egg quality and average egg weight [21,22]. However, diet supplemented with 100 mg/kg chitosan could not obviously change the digestibility of nutrients in 18-week-old layers [12]. Furthermore, diets containing 50 g/kg chitosan had no significant effect on BWG, AFI and FCR in broilers. In addition, high level of chitosan inhibited the growth of broilers [18]. To further illustrated whether chitosan can promote the production performance and antioxidant defense of laying hens and what is the optimal dose range, this study was conducted to evaluate the dietary effects of chitosan on production performance, antioxidant status of serum, liver and duodenum, exploring the molecular mechanism of chitosan regulating the antioxidant function of laying hens preliminarily.

## 2. Materials and Methods

All procedures were approved by the Laboratory Animal Sciences and Technical Committee of the Standardisation Administration of China (SAC/TC281). Besides, the use and care of laboratory animals implemented in accordance with the national standard “Guidelines for Ethical Review of Animal Welfare” (GB/T 35892-2018).

### 2.1. Animals, Experiment Design and Treatments

Chitosan, with the viscosity of 45 mPa/s and the deacetylation degree of 85.09%, was provided by Shandong Haidebei Biotechnology Co., Ltd. (Jinan, China). All of 450 26-week-old, healthy, uniform body weight laying breeder hens (with the strain of Hy-Line brown and HDEP of 92.44% ± 0.030%) were selected in a single-factor completely randomized design. The breeders were placed in wire mesh cages and randomly allotted to 5 treatments, with each treatment comprising 6 equal replicates of 15 birds in a cage (100 × 50 × 50 cm). The laying breeders in five treatments were fed the basal diet supplemented with 0 (control), 250, 500, 1000 or 2000 mg/kg chitosan, respectively. The composition and nutrient level of the control diet is found in Table 1. The experiment lasted 8 weeks, during which time experimental diets and water were available ad libitum for breeders. Before experiment, the poultry facilities and surroundings were fumigated to disinfect by using methanal plus potassium permanganate. Regular immunization and disinfection were performed throughout the experimental period.

### 2.2. Egg Production and Sample Collection

Hen-day egg production (HDEP), average daily feed intake (ADFI) and egg weight were recorded daily. FCR was calculated as ADFI/egg weight. At trial weeks 4 and 8, one laying breeder from each replicate were randomly selected, weighed and slaughtered by cervical dislocation. Samples of serum, liver and duodenum were collected based on the measures as described by Tufarelli et al. (2016) [23]; as well as, samples were stored at −80 °C until analysis.

### 2.3. Antioxidative Enzyme Activities

The MDA content, total antioxidant capacity (T-AOC) and activities of CAT, GSH-Px and T-SOD from serum, liver and duodenum samples were measured using commercial antioxidant kits (Jiancheng Bioengineering Institute, Nanjing, China) according to the instructions of manufacture.

### 2.4. Total RNA Extraction and Quality Determination

Total RNA of liver and duodenum was extracted from 0.5 g tissue, respectively, using the RNAiso Plus Kits (Takara Bio Inc., Kusatsu, Japan) following the specifications. The purity and integrality of the RNA was measured spectrophotometrically using a microplate reader (Thermo Scientific, Waltham, MA, USA) at 260 nm and 280 nm [24]. Reverse transcription was performed by Reverse Transcription System Kits (Takara Bio Inc., Kusatsu, Japan) to obtain cDNA, according to manufacturer’s instructions.

### 2.5. Quantitative Real-Time Polymerase Chain Reaction

The generating cDNAs were used in quantitative real-time polymerase chain reaction (qRT-PCR). *β-actin* was used as a housekeeping gene. The primer sequences of *β-actin*, *GSH-Px, SOD1*, *SOD2*, *CAT*, *thioredoxin reduction enzymes 1*(*TrxR1*) and *Nrf2* were designed by Gene bank database and synthesized by Sangon Biotech (Shanghai, China) (Table 2). The qRT-PCR was performed in 20 μL reactions including 10 μL SYBR Premix Ex Taq TM π (Takara BioInc, Kusatsu, Japan), 7.2 μL of nuclease-free water, 2 μL of cDNA, 0.4 μL of forward primer (10 pmol) and 0.4 μL of reverse primer (10 pmol), using a IQ5 Multicolor Real-Time PCR Detection System (Bio-Rad, Hercules, CA, USA). Each reaction was run in duplicate with the following PCR program: 95 °C for 30 s (initial denaturation) followed by 40 cycles of 95 °C for 30 s (denaturation), 60 °C for 30 s (annealing), 72 °C for 20 s (amplification). The 2^−ΔΔCt^ method was used to analyze the relative quantity of target gene mRNA.

### 2.6. Statistical Analyses

The statistical significance of data was evaluated by SAS 9.0, using ANOVA procedure on normally distributed data, otherwise using Kruskal–Wallis test. Differences among treatment means were analyzed by Tukey-Kramer method. Multivariate regression analysis was used to determine linear and quadratic responses. Differences among the mean values was considered significant at *p* < 0.05.

## 3. Results

### 3.1. Egg Production

Results of laying performance was shown in Table 3, both during week 0~4 (*p* = 0.013) and week 5~8 (*p* = 0.020), dietary chitosan quadratically increased HDEP; however, it had no effect on ADFI, FCR and egg weight in laying breeders. During week 1~4, 500 mg/kg group showed the highest HDEP, and higher (*p* = 0.010) than control. During week 5~8, the HDEP was higher (*p* = 0.009) in 250 and 500 mg/kg group than that in 2000 mg/kg group. During the whole trial period, dietary chitosan supplementation also quadratically increased HDEP (*p* = 0.010), as well as, the 500 mg/kg group showed the optimum effect. In addition, dietary chitosan supplementation linearly increased ADFI (*p* = 0.043) and FCR (*p* = 0.039).

### 3.2. Antioxidative Activities in Serum

The results of antioxidant enzyme activities in serum were shown in Table 4. Throughout the trial phase, serum T-SOD activity increased quadratically (*p* = 0.016). Besides, it was highest in the 500 mg/kg group. The concentration of MDA decreased quadratically (*p* = 0.026) with the increase of chitosan. During week 5~8, anti-hyperoxide anionic capacity increased quadratically (*p* = 0.004); however, inhibit hydroxyl radical ability increased linearly (*p* = 0.020). From week 1 to 4, the activity of GSH-Px was increased (*p* = 0.035) at the chitosan level of 500 mg/kg; as well as, it was also increased (*p* = 0.026) at the level of 250 mg/kg from week 5 to 8.

### 3.3. Antioxidative Activities and Gene Expressions in Liver

The results of antioxidant enzyme activities in liver were shown in Table 5; as well as, the corresponding gene expressions were shown in Table 6. During week 1~4, chitosan quadratically increased (*p* < 0.05) the activity of T-SOD, anti-hyperoxide anionic capacity and the gene expression of *SOD1*, *SOD2* and *Nrf2*. When diet containing 500 mg/kg chitosan, above variables reached the maximum. Among them, T-SOD activity was higher (*p* = 0.008) in 500 mg/kg group than that in 2000 mg/kg group. Anti-hyperoxide anionic capacity was higher (*p* = 0.040) in 500 mg/kg group than other groups except for 1000 mg/kg. The expression of *SOD2* was higher (*p* = 0.015) in 500 mg/kg group than contrast. During week 5~8, chitosan linearly decreased (*p* = 0.002) MDA concentration, but quadratically increased (*p* < 0.05) the activity of CAT, T-SOD and TrxR and the expression of *CAT* and *GSH-Px*. All above variables reached the maximum at the chitosan level of 500 mg/kg. Among them, CAT activity and anti-hyperoxide anionic capacity were higher (*p* < 0.05) in 250~1000 mg/kg group than that in the other two groups. As well as, T-SOD activity was higher (*p* = 0.001) in 500 mg/kg group than 1000 and 2000 mg/kg group. The *GSH-Px* expression was higher (*p* = 0.016) in 500 and 1000 mg/kg group than that in contrast and 2000 mg/kg group. The *SOD2* expression was higher (*p* = 0.013) in 500 mg/kg than that in contrast and 2000 mg/kg group. 

### 3.4. Antioxidative Activities and Gene Expressions in Duodenum

The results of antioxidant enzyme activities in duodenum were shown in Table 7, the corresponding gene expressions were shown in Table 8. During week 1~4, with increasing dosage of chitosan, the activity of GSH-Px, T-SOD and anti-hyperoxide anionic capacity increased quadratically (*p* < 0.05). In particular, the activity of T-SOD in 500 mg/kg group was higher (*p* = 0.025) than that in control. Besides, anti-superoxide anion capacity in 500 mg/kg and 1000 mg/kg group was higher (*p* = 0.018) than that in 0 and 2000 mg/kg group. During week 5~8, with increasing dosage of chitosan, the activity of T-SOD, inhibit hydroxyl radical ability and anti-hyperoxide anionic capacity increased quadratically (*p* < 0.05). The activity of T-SOD in 250~1000 mg/kg group was higher (*p* = 0.019) than that in other two group. The activity of TrxR in 250~1000 mg/kg group was higher (*p* = 0.030) than control. The anti-hyperoxide anionic capacity was higher (*p* = 0.042) in 500 mg/kg and 1000 mg/kg group than control. During week 1~4, with increasing dosage of chitosan, *SOD1*, *SOD2* and *TrxR1* gene expression increased quadratically (*p* < 0.05). The gene expression of *SOD2* in the 500 mg/kg group was also higher (*p* < 0.05) than that in 2000 mg/kg group. During week 5~8, with the increasing dosage of chitosan, *GSH-Px*, *SOD2* and *TrxR1* gene expression increased quadratically (*p* < 0.05). The gene expression of *GSH-Px* was up-regulated (*p* = 0.012) in the chitosan level of 500 mg/kg and 1000 mg/kg than that in control. The *SOD2* expression was highest in the level of 250 mg/kg, and higher (*p* = 0.007) than that in other dosages, among which *SOD2* expression was higher in the level of 500 mg/kg and 1000 mg/kg than the other two groups. The *TrxR1* expression showed similar effect to the *SOD2* expression; however, the *TrxR1* expression was highest in the level of 500 mg/kg. It is noteworthy that the expression levels of above genes were all higher than those of the control.

## 4. Discussion

### 4.1. Egg Production

HDEP, ADFI, average egg weight and FCR are important indicators reflecting the performance of laying hens. Results of early studies on the use of chitosan dietary supplementation showed that feeding broilers or laying hens an amount below 1.4 g/(kg·BW) per day is harmless, which could reduce the serum concentrations of cholesterol, triglycerides and free fatty acids in birds fed a cholesterol-additive diet [25]. The doses used in this experiment were all below 1.4 g/(kg·BW) per day. Our study revealed that addition of chitosan could promote HDEP and FCR, as well as, the effect was better at the level of 250 mg/kg and 500 mg/kg. It may be related to the effect of chitosan to improve birds antioxidation and bone biomechanical indicators. The results of this experiment showed that chitosan could increase the activity of antioxidant enzymes and related gene expression in blood, liver and duodenum of laying hens to different degrees. Another study indicated that diets containing different levels of chitosan could reduce the oxidation ability of egg yolk and improve the antioxidant performance of plasma in laying hens [26]. Moreover, Sylwester et al. (2018) demonstrated adding 100 mg/kg chitosan to the diet significantly improved the fracture strength, bending load and mineralization of the femur [12].

Li et al. (2019) reported that broiler diet supplemented with 30 mg/kg chitooligosaccharide could improve the FCR [11]. This was consistent with our findings. Swiatkiewicz et al. (2013) found that dietary including 100 mg/kg chitosan with high level of distillers dried grains with solubles (DDGS) improved the digestibility of nutrients during the entire feeding period and promoted the deposition of nitrogen and calcium, thereby improving the production performance of laying hens [20]. This indicated that perhaps the addition of low-dose chitosan is more conducive to production performance of poultry. However, the resent research on layers showed diet supplementation with 100 mg/kg chitosan could not significantly affect the digestibility of dry matter, organic matter, crude fat, nitrogen-free leachate, crude fiber and ash in 18-week-old layers [12]. In this study, the dosage of chitosan was not lower than 250 mg/kg. Therefore, the appropriate dose and mechanism of chitosan diet to promote the production performance of laying hens needs further research and discussion.

### 4.2. Antioxidative Activities in Serum

In the process of external stress and nutrient digestion and metabolism, animal organism is prone to produce free radicals, including hydroxyl free radicals, hydrogen peroxide, superoxide anion and lipid peroxides [27]. The activities of CAT, GSH-Px, T-SOD, T-AOC and TrxR serve as protective responses to eliminate reactive free radicals [28,29]. MDA is the final product of lipid peroxidation, and its content can reflect the degree of lipid peroxidation in the body. The ability to inhibit hydroxyl radicals and the ability to resist superoxide anion can reflect the body’s own redox state. Early studies have shown that chitosan and its derivatives have a strong ability to scavenge free radicals, such as hydroxyl radicals and superoxide anions [30]. The results of previous studies showed that chitosan could reduce the concentration of active oxygen and the degree of lipid peroxidation in animals by regulating the activity of related antioxidant defense enzyme systems, thereby protecting the body from the damage of oxides. Farivar et al. (2018) added 0, 200, 400, 800 and 1600 ppm chitosan to the diet of laying hens, and the results of their study showed that the plasma antioxidant capacity increased linearly with the increase of chitosan levels in the diet [26]. The present study found that, with increasing dosage of chitosan, the activity of GSH-Px and T-SOD increased quadratically, MDA content decreased quadratically. When diet supplemented with 500 mg/kg chitosan, the serum antioxidant status in laying breeders was obviously improved. The 1000 mg/kg group also showed a certain promotion effect; however, when 2000 mg/kg chitosan was added, there seemed to be no promotion.

### 4.3. Antioxidative Activities and Gene Expressions in Liver and Duodenum

Egg breeders are susceptible to oxidative stress during peak egg production. The liver, as its main metabolic organ, easily produces free radicals, which can cause oxidative damage and increase the susceptibility of poultry to diseases. One study in mice showed that addition of 200 mg/kg chitooligosaccharide to the diet of oxidative stress mice caused by hydrogen peroxide could relatively increase the activities of SOD, CAT, GSH-Px and T-AOC in serum, liver, spleen and kidney, as well as relatively reduce the concentration of MDA, thereby alleviating the oxidative damage caused by H_2_O_2_ [31]. There are some similar reports, chitosan can relieve oxidative stress in the liver of mice by up-regulating the gene expression of antioxidant enzymes and decreasing pro-inflammatory cytokines, neutrophils infiltration and macrophage polarization, thereby improving non-alcoholic fatty liver caused by high-fat diets [32]. Mosaad et al. (2017) pointed out that after feeding male rats with chitosan nanoparticles, the expression levels of *GSH-Px* and *SOD1* genes in kidney tissues were up-regulated [33]. Similar results were found in the current study. Chitosan could regulate the activities of T-SOD, CAT, GSH-Px and TrxR and the expression level of related genes in liver tissue in a dose-dependent manner. In the current study, compared with week 1~4, the effect of dietary chitosan on liver antioxidant status was more significant at week 5~8. Moreover, 500 mg/kg group showed a better expression promoting effect, but the promoting effect was diminished when the diet containing 2000 mg/kg chitosan.

The duodenum is a part of the small intestine, and is one of the important structures connected with the external environment. It has a large workload and a high oxidative metabolic rate, resulting in rich active oxygen content and being susceptible to oxidative damage. A study showed that chitosan could promote the absorption of certain small molecules, peptides and nutrients in the small intestine, which is beneficial to intestinal health [34,35]. Meanwhile, it has also been documented that low molar mass chitosan has ability to adsorb active oxygen radicals, and thus it is used as a potential antioxidant [36]. Li et al. (2017) pointed that addition of chitosan to the diet enhanced the activities of T-AOC, GSH-Px and T-SOD, and reduced the content of MDA in broiler ileal mucosa [10]. Similar results were also reported by Li et al. (2019) [11]. Up to now, scarce published papers have concentrated on the ameliorative effects of chitosan on duodenal antioxidant status of laying breeders. In the present study, with the increase of chitosan dosage, the activity of T-SOD, the inhibit hydroxyl radical ability and the anti-hyperoxide anionic capacity showed a significant increase quadratically. In addition, 250~500 mg/kg groups showed better promotion effects, and the promotion effect was weakened at the level of 2000 mg/kg. These results suggested that the appropriate amount of chitosan could increase the antioxidant status of duodenal tissue; however, the promotion effect was weakened at high dose of chitosan.

### 4.4. Nrf2 Gene Expressions in Liver and Duodenum

The relative expression of antioxidant enzyme-related genes in the body is simultaneously regulated by multiple signaling pathways, among which Nrf2 is one of the most important pathways. Nrf2 is a redox-sensitive transcription factor regulating ARE, which regulate the expression of Nrf2-mediated phase detoxifying enzyme genes [13,37]. Chitosan has a protective effect on liver cells injured by hydrogen peroxide, and mainly induces the expression of antioxidant enzymes by mediating Nrf2 translocation nuclei [38]. Chitosan oligosaccharide could also up-regulate the gene expression of hepatic antioxidant enzymes by activating the nuclear factor Nrf2 pathway, and relieve oxidative stress in the liver of mice [32]. A recent study also showed that chitooligosaccharide increased jejunal occludin and ileal *claudin 2*, *Nrf2* and *HO-1* expression, decreased jejunal *interferon-γ* and *interferon-β* abundance, and then alleviate LPS-induced intestinal barrier damage, and immunological and oxidative stress in laying hens [39]. Similar results were found in the present study. In this study, chitosan dietary could quadratically up-regulate the expression of *Nrf2* gene in liver tissue. Similar with earlier reports of Ahn et al. (2017) [38], we also observed diet supplemented with 500 mg /kg chitosan improved *Nrf2* expression, but as the increase of chitosan dosage, the promotion effect was weakened at the level of 2000 mg/kg. Combined with the results of antioxidant enzyme gene expression, it is suggested that chitosan might regulate the downstream antioxidant enzyme gene expression through the Nrf2 signaling pathway, thereby enhancing antioxidant enzyme activities and protecting the body from lipid peroxide.

It can be seen from the results of our study, diet supplemented with chitosan had a significant effect on the serum, liver and duodenal antioxidant enzyme activities of laying breeders, and its effect is dose-dependent. Different dietary levels of chitosan (250~1000 mg/kg) improved the antioxidant status of serum, liver and duodenal, while the promotion effect was weakened even inhibited at the level of 2000 mg/kg in the diet. Regarding to the mechanism by which chitosan enhances the antioxidant function of layer breeders, Chou et al. (2003) pointed out that chitosan inhibited LPS-damaged macrophages from producing excessive amounts of arachidonic acid and prostaglandin E2, thereby reducing the production of pro-inflammatory factors, and improving the antioxidant function of cells [40]. However, another study has also shown that the activation of the Nrf2 pathway is related to the inhibition of inflammation [41]. Thus, it is speculated that the promotion effect of chitosan on the antioxidant function of laying breeders might be related to the change of secretion level of antioxidant related hormones, or it might be related to the inhibition of inflammatory response. Moreover, Nrf2 plays an important role in maintaining the redox homeostasis of liver, which can improve the defense ability of oxidative stress by activating the transcription of antioxidant genes [42]. Combined with the results of the current study, it is speculated that the antioxidative mechanism of chitosan in laying hens was likely to that chitosan activated the Nrf2 pathway and up-regulates the upstream *Nrf2* gene expression, thereby promoting the expression of downstream phase II detoxifying enzyme genes and then increasing their activities, and ultimately improving the antioxidant capacity of laying breeders.

Regarding to the weakening promotion effect or even no effect of 2000 mg/kg group on antioxidant capacity, it might be related to the bacteriostatic effect of chitosan. It is reported that chitosan could inhibit the activity of fungi, Gram-positive bacteria and Gram-negative bacteria [43]. Moreover, when the chitosan concentration was higher, the protonated chitosan could be wrapped on the surface of bacterial cells which not only prevented the extravasation of constituents in bacterial cells, but also made them repelled each due to the positive charge, thereby preventing its agglutination [44]. Our study did not evaluate the effect of chitosan on the gut microbial community of laying hens, and the mechanism needs further verification.

## 5. Conclusions

Dietary chitosan supplementation increased HDEP and FCR, and the effect was better at the level of 250~500 mg/kg; however, chitosan had no prominent effect on feed intake and average egg weight. Supplementation of chitosan could dose-dependently promote the antioxidant status in serum, liver and duodenum of layer breeders. As well as, it has a better promotion effect at the level of 500 mg/kg, and the effect was weakened at the level of 2000 mg/kg. Chitosan was likely to enhance the gene expression and activities of Nrf2-mediated phase II detoxification enzymes by up-regulating the expression of *Nrf2*, thereby improving the antioxidant capacity of laying breeders.

## Figures and Tables

**Table 1 animals-12-01225-t001:** Composition and nutrient levels of basal diet (air-dry basis, %).

Composition	Content (%)	Nutritional Level	Content (%)
Corn	62.70	ME (MJ/kg)	11.09
Soybean meal	26.30	CP	16.61
Limestone	8.50	Ca	3.50
Met	0.10	P	0.35
Bone	1.00	Met	0.42
Choline chloride	0.10	Lys	0.85
Salt	0.30	Trp	0.21
Premix ^a^	1.00		
Total	100.00		

Note: ^a^ Premix could provide the following based on per kilogram diet: Vitamin A 751 IU, Vitamin D 755 IU, Vitamin E 8.8 IU, Vitamin K 2.2 mg, Vitamin B1 10.55 mg, Vitamin B6 24.41 mg, Vitamin B12 120.01 mg, nicotinic acid 19.8 mg, folic acid 0.28 mg, Mn 50 mg, Fe 25 mg, Cu 2.5 mg, Zn 50 mg, I 1.0 mg and Se 0.15 mg.

**Table 2 animals-12-01225-t002:** Sequence of the object primers.

Gene	GeneBank No.	Primer Sequence	Length/bp
*β-actin*	NM_205518	F:ATCCGGACCCTCCATTGTC	120 bp
		R:AGCCATGCCAATCTCGTCTT	
*GSH-Px*	NM_204220	F:CATCACCAACGTGGCGTCCAA	92 bp
		R:GCAGCCCCTTCTCAGCGTATC	
*SOD1*	NM_205064	F:TTGTCTGATGGAGATCATGGCTTC	98 bp
		R:TGCTTGCCTTCAGGATTAAAGTGAG	
*SOD2*	NM_204211	F:CAGATAGCAGCCTGTGCAAATCA	86 bp
		R:GCATGTTCCCATACATCGATTCC	
*CAT*	NM_001031215.1	F:ACCAAGTACTGCAAGGCGAAAGT	91 bp
		R:ACCCAGATTCTCCAGCAACAGTG	
*TrxR1*	NM_00103076.2	F:TACGCCTCTGGGAAATTCGT	114 bp
		R:CTTGCAAGGCTTGTCCCAGTA	
*Nrf2*	NM205117.1	F:GATGTCACCCTGCCCTTAG	143 bp
		R:CTGCCACCATGTTATTCC	

Note: F = Forward primer; R = Reverse primer.

**Table 3 animals-12-01225-t003:** Effects of dietary chitosan on the laying performance in laying breeders.

Items	Levels of Chitosan (mg/kg)					Sign.	SEM	*p*-Value	
	0	250	500	1000	2000			Linear	Quadratic
1–4 week									
HDEP/%	90.52 ^b^	94.14 ^ab^	95.19 ^a^	93.06 ^ab^	93.3 ^ab^	0.010	0.973	0.116	0.013
ADFI/g	121.88	120.05	126.43	125.57	124.43	0.596	3.157	0.853	0.902
Egg weight/(g/d)	55.10	55.63	54.65	54.85	55.68	0.089	0.304	0.412	0.115
FCR	2.24	2.15	2.29	2.28	2.26	0.470	0.062	0.632	0.508
5–8 week									
HDEP/%	90.87 ^ab^	94.76 ^a^	94.35 ^a^	92.98 ^ab^	90.13 ^b^	0.009	0.997	0.121	0.020
ADFI/g	123.40	123.30	127.50	129.00	130.00	0.158	2.216	0.902	0.495
Egg weight/(g/d)	56.57	56.77	56.10	56.34	55.68	0.138	0.305	0.250	0.053
FCR	2.19	2.17	2.27	2.28	2.32	0.103	0.065	0.576	0.274
The whole period									
HDEP/%	90.70 ^b^	94.45 ^a^	94.77 ^a^	93.02 ^ab^	91.72 ^b^	0.003	0.701	0.424	0.010
ADFI/g	122.64	121.67	126.96	127.28	127.22	0.112	1.908	0.043	0.052
Egg weight/(g/d)	55.84	56.20	55.37	55.59	55.68	0.123	0.279	0.480	0.507
FCR	2.22 ^ab^	2.17 ^b^	2.28 ^ab^	2.29 ^ab^	2.30 ^a^	0.028	0.034	0.039	0.077

^a,b^ Means within the same row not followed by the same letters are significantly different at *p* < 0.05.

**Table 4 animals-12-01225-t004:** Effect of dietary chitosan supplementation on serum antioxidant variables of laying breeders.

Items	Levels of Chitosan (mg/kg)					Sign.	SEM	*p*-Value	
	0	250	500	1000	2000			Linear	Quadratic
1–4 week									
GSH-Px (nmol/mL)	231.46 ^b^	251.57 ^ab^	255.65 ^a^	235.64 ^ab^	236.68 ^ab^	0.035	6.289	0.538	0.501
MDA (nmol/mL)	8.93 ^a^	7.22 ^ab^	7.13 ^ab^	7.11 ^ab^	7.26 ^ab^	0.015	0.413	0.139	0.026
CAT (U/mL)	6.44	8.75	7.02	7.01	7.16	0.345	0.570	0.938	0.899
T-SOD (U/mL)	123.33 ^b^	127.06 ^ab^	134.45 ^a^	130.49 ^ab^	128.78 ^ab^	0.028	2.503	0.292	0.016
T-AOC (U/mL)	7.47	10.65	8.68	8.67	8.17	0.117	0.692	0.595	0.631
Inhibit hydroxyl radical ability (U/mL)	61.29	65.75	69.01	62.84	62.05	0.193	2.865	0.652	0.514
Anti-hyperoxide anionic capacity (U/L)	341.45	345.39	341.23	350.66	344.74	0.958	9.179	0.873	0.777
5–8 week									
GSH-Px (nmol/mL)	211.99 ^b^	243.76 ^a^	227.10 ^ab^	221.24 ^ab^	220.68 ^ab^	0.026	5.107	0.616	0.765
MDA (nmol/mL)	9.47 ^a^	8.51 ^ab^	7.33 ^c^	7.21 ^c^	7.82 ^c^	0.009	0.438	0.002	0.001
CAT (U/mL)	7.23	7.38	7.31	7.64	7.59	0.989	0.431	0.649	0.884
T-SOD (U/mL)	121.37 ^b^	126.41 ^ab^	132.33 ^a^	128.02 ^ab^	123.58 ^b^	0.005	2.446	0.986	0.039
T-AOC (U/mL)	8.85	8.69	9.29	8.77	8.77	0.838	2.449	0.955	0.996
Inhibit hydroxyl radical ability (U/mL)	67.49	73.99	76.94	74.00	73.91	0.189	2.472	0.020	0.119
Anti-hyperoxide anionic capacity (U/L)	327.19 ^b^	372.37 ^a^	375.00 ^a^	375.26 ^a^	346.84 ^ab^	0.033	9.696	0.875	0.004

^a,b,c^ Means within the same row not followed by the same letters are significantly different at *p* < 0.05.

**Table 5 animals-12-01225-t005:** Effect of dietary chitosan supplementation on liver antioxidant variables of laying breeders.

Items	Levels of Chitosan (mg/kg)					Sign.	SEM	*p*-Value	
	0	250	500	1000	2000			Linear	Quadratic
1–4 week									
GSH-Px (U/mg protein)	6.56	7.97	7.97	6.81	7.33	0.277	0.552	0.715	0.332
MDA (nmol/mg protein)	3.88	2.58	4.04	3.62	3.47	0.759	0.259	0.112	0.126
CAT (U/mg protein)	16.08	15.83	16.58	16.01	16.31	0.945	0.391	0.122	0.293
T-SOD (U/mg protein)	652.58 ^bc^	696.96 ^ab^	728.46 ^a^	701.52 ^ab^	635.15 ^c^	0.008	18.826	0.021	0.001
T-AOC (U/mg protein)	2.12	2.40	2.17	2.12	2.13	0.953	0.237	0.125	0.16
TrxR (U/mg protein)	49.57	61.89	61.02	59.26	56.08	0.323	4.645	0.745	0.312
Inhibit hydroxyl radical ability (U/mg protein)	16.15	17.82	17.62	18.14	17.7	0.848	0.651	0.254	0.555
Anti-hyperoxide anionic capacity (U/mg protein)	58.45 ^ab^	55.75 ^b^	72.97 ^a^	62.59 ^ab^	56.80 ^b^	0.040	4.176	0.432	0.002
5–8 week									
GSH-Px (U/mg protein)	7.35 ^b^	9.61 ^ab^	10.45 ^a^	7.96 ^ab^	9.01 ^ab^	0.045	0.685	0.861	0.675
MDA (nmol/mL)	3.13 ^ab^	2.38 ^b^	2.89 ^ab^	3.12 ^ab^	3.25 ^a^	0.024	0. 189	0.002	0.006
CAT (U/mg protein)	15.84 ^b^	18.78 ^a^	19.38 ^a^	18.90 ^a^	16.79 ^b^	0.005	0.611	0.001	<0.001
T-SOD (U/mg protein)	771.50 ^abc^	802.59 ^ab^	846.32 ^a^	730.85 ^c^	720.34 ^c^	0.001	23.709	0.001	0.001
T-AOC (U/mg protein)	1.41	1.72	1.59	1.27	1.31	0.371	0.196	0.239	0.409
TrxR (U/mg protein)	95.94	105.91	107.39	94.86	82.32	0.123	7.400	0.026	0.046
Inhibit hydroxyl radical ability (U/mg protein)	17.62	20.39	23.60	17.25	15.87	0.095	2.014	0.665	0.302
Anti-hyperoxide anionic capacity (U/mg protein)	39.83 ^b^	55.36 ^a^	56.84 ^a^	55.68 ^a^	44.80 ^b^	0.008	3.925	0.005	0.001

^a,b,c^ Means within the same row not followed by the same letters are significantly different at *p* < 0.05.

**Table 6 animals-12-01225-t006:** Effect of dietary chitosan supplementation gene expression of antioxidant enzymes in liver.

Items	Levels of Chitosan (mg/kg)					Sign.	SEM	*p*-Value	
	0	250	500	1000	2000			Linear	Quadratic
1–4 week									
*GSH-Px*	1.04	1.15	1.25	1.11	1.14	0.058	0.069	0.503	0.227
*CAT*	1.02	1.36	1.46	1.38	1.27	0.239	0.160	0.576	0.194
*SOD1*	1.01	1.28	1.52	1.38	1.13	0.145	0.135	0.988	0.032
*SOD2*	1.03 ^b^	1.52 ^ab^	1.87 ^a^	1.55 ^ab^	1.35 ^ab^	0.015	0.221	0.527	0.055
*TrxR1*	1.04	1.35	1.23	1.16	1.09	0.081	0.095	0.676	0.482
*Nrf2*	1.05	1.17	1.34	1.31	1.09	0.223	0.088	0.589	0.015
5–8 week									
*GSH-Px*	1.02 ^b^	1.68 ^ab^	1.95 ^a^	1.78 ^a^	1.20 ^b^	0.016	0.219	0.683	0.001
*CAT*	1.02	1.15	1.83	1.63	1.01	0.098	0.265	0.920	0.027
*SOD1*	1.03	1.09	1.65	1.4	1.27	0.086	0.179	0.434	0.084
*SOD2*	1.03 ^b^	1.45 ^ab^	2.15 ^a^	1.36 ^ab^	1.14 ^b^	0.013	0.245	0.869	0.088
*TrxR1*	1.06	1.37	1.32	1.27	1.27	0.323	0.11	0.369	0.087
*Nrf2*	1.08	1.15	1.39	1.23	1.17	0.118	0.081	0.500	0.094

^a,b^ Means within the same row not followed by the same letters are significantly different at *p* < 0.05.

**Table 7 animals-12-01225-t007:** Effect of dietary chitosan supplementation on duodenum antioxidant variables of laying breeders.

Items	Levels of Chitosan (mg/kg)					Sign.	SEM	*p*-Value	
	0	250	500	1000	2000			Linear	Quadratic
1–4 week									
GSH-Px (U/mg protein)	12.75	18.89	19.28	20.29	17.91	0.119	1.835	0.244	0.040
MDA (nmol/mg protein)	5.63	4.29	3.92	4.83	4.59	0.419	0.564	0.877	0.620
CAT (U/mg protein)	5.24	5.74	6.99	6.92	6.20	0.601	0.728	0.493	0.357
T-SOD (U/mg protein)	354.87 ^c^	428.67 ^ab^	478.32 ^a^	378.33 ^bc^	373.27 ^bc^	0.025	20.751	0.193	0.013
T-AOC (U/mg protein)	2.24	2.25	2.61	2.99	2.40	0.500	0.273	0.65	0.231
TrxR (U/mg protein)	43.37	45.52	54.64	44.28	40.39	0.117	3.207	0.292	0.055
Inhibit hydroxyl radical ability (U/mg protein)	41.01	48.82	59.73	47.15	45.93	0.098	4.263	0.702	0.054
Anti-hyperoxide anionic capacity (U/mg protein)	92.47 ^b^	129.87 ^a^	134.60 ^a^	137.38 ^a^	84.77 ^b^	0.018	11.31	0.681	0.004
5–8 week									
GSH-Px (U/mg protein)	12.83	18.04	21.78	16.13	12.94	0.065	2.136	0.298	0.093
MDA (nmol/mL)	4.46	4.89	4.37	4.52	4.37	0.899	0.445	0.738	0.945
CAT (U/mg protein)	7.28	7.05	5.81	8.38	7.45	0.224	0.631	0.693	0.627
T-SOD (U/mg protein)	344.81 ^b^	406.01 ^a^	419.95 ^a^	413.47 ^a^	346.36 ^b^	0.019	16.945	0.793	0.005
T-AOC (U/mg protein)	1.85	1.90	2.14	1.43	1.25	0.181	0.228	0.181	0.231
TrxR (U/mg protein)	43.43 ^b^	55.36 ^a^	55.17 ^a^	53.12 ^a^	50.35 ^ab^	0.030	3.174	0.497	0.075
Inhibit hydroxyl radical ability (U/mg protein)	44.34	46.79	66.55	57.38	46.98	0.134	4.175	0.745	0.040
Anti-hyperoxide anionic capacity (U/mg protein)	113.24 ^b^	162.32 ^ab^	168.45 ^a^	164.10 ^a^	158.24 ^ab^	0.042	9.616	0.198	0.011

^a,b,c^ Means within the same row not followed by the same letters are significantly different at *p* < 0.05.

**Table 8 animals-12-01225-t008:** Effect of dietary chitosan supplementation gene expression of antioxidant enzymes in duodenum.

Items	Levels of Chitosan (mg/kg)					Sign.	SEM	*p*-Value	
	0	250	500	1000	2000			Linear	Quadratic
1–4 week									
*GSH-Px*	1.05	1.28	1.98	1.34	1.21	0.058	0.210	0.928	0.091
*CAT*	1.01	1.14	1.31	1.15	1.04	0.325	0.124	0.886	0.403
*SOD1*	1.03 ^b^	1.56 ^a^	1.87 ^a^	1.51 ^ab^	1.35 ^ab^	0.020	0.159	0.790	0.033
*SOD2*	1.11 ^c^	1.35 ^ab^	1.90 ^a^	1.23 ^bc^	1.05 ^c^	0.005	0.182	0.346	0.036
*TrxR1*	1.06 ^c^	1.40 ^abc^	1.69 ^a^	1.60 ^ab^	0.98 ^bc^	0.022	0.223	0.497	0.009
*Nrf2*	1.10	1.24	1.45	1.33	1.25	0.270	0.118	0.371	0.054
5–8 week									
*GSH-Px*	1.06 ^b^	1.55 ^ab^	2.29 ^a^	2.39 ^a^	1.66 ^ab^	0.012	0.343	0.297	0.001
*CAT*	1.05	1.15	1.50	1.28	1.25	0.423	0.134	0.322	0.154
*SOD1*	1.00	1.41	1.64	1.41	1.29	0.078	0.163	0.620	0.087
*SOD2*	1.12 ^c^	2.11 ^a^	1.61 ^b^	1.58 ^b^	1.15 ^c^	0.007	0.224	0.297	0.046
*TrxR1*	1.09 ^c^	2.27 ^b^	3.06 ^a^	1.51 ^b^	1.08 ^c^	0.012	0.418	0.210	0.044
*Nrf2*	1.02	1.29	1.41	1.30	1.24	0.186	0.123	0.507	0.057

^a,b,c^ Means within the same row not followed by the same letters are significantly different at *p* < 0.05.

## Data Availability

The data presented in this study are available on request from the corresponding author.

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
