# Peer review of "Dietary Chitosan Supplementation Improved Egg Production and Antioxidative Function in Laying Breeders"

_animals, 2022, doi:10.3390/ani12101225_

Round 1

Reviewer 1 Report

In this study, the authors investigated the effects of dietary chitosan on egg production and antioxidant parameter. The aim and methods of this study are straightforward. However, there are some points which should be revised/considered.

Major comments;

  1.  Duncan's multiple range test is not recommended for multiple comparison. Other methods, for example Tukey-Kramer method and Dunnett's test should be used.
  2. The authors use the following expressions in Results section; "increased quadratically" and "increased linearly". Maybe the authors used these expressions based on the results of regression analysis. However, these expressions are not correct to show the results. For example, significant differences are observed in serum T-SOD activity, but the numerical data  (123.33, 127.06, 134.45, 130.49, and 128.78 U/mL) is not increased quadratically.

Minor comments;

  1.  The authors use "growth performance" in the title and subsection (2.3). But, in this study, they evaluated egg production but growth performance. It appears to be better to change "growth performance" to "egg production".
  2. Spell out "ADFI" (Line 125).
  3. Calculation method of egg production should be written.
  4. PCR program does not  appear to be correctly described (Line 154-157).

Reviewer 2 Report

Dear Authors,

Thank you for submitting this interesting paper investigating the use of chitosan supplementation for chicken broiler diets. Overall, this is an useful study with some potential implications for those working in the poultry industry.

At current however, there seem to be some large revisions required in the manuscript to ensure the work is scientifically robust. I have attached the PDF version of the manuscript with specific comments. Additionally, please consider the following points: 

  1. Proof reading. Many parts of the work are written in poor English, and the scientific value of the work suffers as a result. I would strongly suggest a full proof read is due, ideally by a native english speaker.
  2. Choice of statistical test. ANOVA is a parametric test and assumes normal distribution of data. Did you test your data for normality and if so what was the finding? Please report it here. If the data are not parametric an alternative test (such as Kruskal Wallis or Friedman's ANOVA) would be more appropriate. It is highly unlikely that all tests were normally distributed, so test choice needs to be considered carefully.
  3. Statistical trends. Be careful with interpreting p=0.1 as a statistical trend as you have a 1 in 10 chance that this occurred as a result of chance. It is better to remove any consideration relating to these trends as it means you are increasing your chances of a false positive test result.

With these revisions, the paper should be in a better position for consideration.

Round 2

Reviewer 1 Report

The authors have not changed "growth performance" in the title. Again, I recommend the change. 

Reviewer 2 Report

Dear Authors,

Many thanks for submitting this revised version of the manuscript for review. You have taken into account the feedback provided on the initial review of the paper. The developments to the manuscript have resulted in a more robust paper overall. In light of the revisions, the paper is now in a much better position for consideration.
